# Optimized Random Forest for Solar Radiation Prediction Using Sunshine Hours

**DOI:** 10.3390/mi13091406

**Published:** 2022-08-27

**Authors:** Cesar G. Villegas-Mier, Juvenal Rodriguez-Resendiz, José Manuel Álvarez-Alvarado, Hugo Jiménez-Hernández, Ákos Odry

**Affiliations:** 1Facultad de Informática, Universidad Autónoma de Queretaro, Queretaroo 76230, Mexico; 2Facultad de Ingeniería, Universidad Autónoma de Queretaro, Queretaro 76010, Mexico; 3Department of Control Engineering and Information Technology, University of Dunaújváros, 2400 Dunaújváros, Hungary

**Keywords:** random forest, solar radiation, hyperparameter optimization, neural networks, prediction

## Abstract

Knowing exactly how much solar radiation reaches a particular area is helpful when planning solar energy installations. In recent years the use of renewable energies, especially those related to photovoltaic systems, has had an impressive up-tendency. Therefore, mechanisms that allow us to predict solar radiation are essential. This work aims to present results for predicting solar radiation using optimization with the Random Forest (RF) algorithm. Moreover, it compares the obtained results with other machine learning models. The conducted analysis is performed in Queretaro, Mexico, which has both direct solar radiation and suitable weather conditions more than three quarters of the year. The results show an effective improvement when optimizing the hyperparameters of the RF and Adaboost models, with an improvement of 95.98% accuracy compared to conventional methods such as linear regression, with 54.19%, or recurrent networks, with 53.96%, without increasing the computational time and performance requirements to obtain the prediction. The analysis was successfully repeated in two different scenarios for periods in 2020 and 2021 in Juriquilla. The developed method provides robust performance with similar results, confirming the validity and effectiveness of our approach.

## 1. Introduction

Solar radiation (SR) is the primary source of energy in the world, affected only by the atmosphere, biosphere, and hydrosphere [1]. It has an important impact on the global scale; minor changes in SR trigger considerable changes in Earth’s weather [2], directly affecting global temperatures. The most affected is the sea, resulting in corresponding extreme phenomena such as the *El Niño-Southern Oscillation* [3,4]. Therefore, accurate observations and analyses of both the temporal and spatial variability of SR are essential in research on solar energy, building materials, and extreme weather and climate events [2,5].

In the last couple of years new techniques and algorithms have been developed to predict SR, including traditional models such as empirical models and theoretical parameter-based approaches as well as newer models that apply machine learning (ML) and artificial intelligence (IA) tools, using either data from weather stations on the ground or data collected from weather satellites [6,7].

Sunshine hours are essential because their duration directly affects the amount of solar radiation reaching the land surface. This can be seen directly in the seasons of the year where the length of the day is affected; in the summer, the sunshine hours tend to be much longer than in the winter. The shape of the earth affects the amount of light that reaches the surface. Places near the equator receive the sun’s rays vertically and directly, while at the poles the planet is tilted on its axis of rotation 23${}^{{}^{\circ}}$, and the surface receives much less light. The rotational motion is responsible for these variations; on clear days, the maximum amount of radiation is reached at noon.

Angstrom et al. [8] first proposed the empirical A-P solar radiation prediction model, and were first to establish a linear relationship between global solar radiation and sunshine hours. This approach and its variants are widely used to estimate SR in different parts of the world [6,8,9,10,11], as the models are very simple.

Today, with the constant development of artificial intelligence and machine learning tools, many researchers have begun investigating and implementing these models to predict climatological variables, including solar radiation. An example of this is artificial neural networks (ANN), which play a fundamental role because they work in nonlinear time series, and solar radiation is considered a nonlinear variable. They are used in models where it is impossible to obtain data due to the lack of weather stations.

For the selected study area, which is the city of Queretaro, very few works have investigated solar radiation prediction, leaving a great opportunity to learn about new advances in the subject and apply them in this area. A key objective of this work is to propose adjustments to the input parameters or hyperparameters of machine learning algorithms that are traditionally set by default, which affects performance when generating a prediction. Thus, we seek to create a simple function to optimize them automatically.

Prior work has been carried out to predict solar radiation using traditional ML algorithms. However, this work considered the sunshine hours available during the day and considered variables such as temperature, humidity of the zone (Querearo), and atmospheric pressure. The presented model helps to automatically adjust the hyperparameters, achieving an improvement in the time needed for prediction without incurring a very high computational cost.

This paper aims to improve the RF algorithm using a simple fitting algorithm to find the best input parameters, in turn using a regression analysis of solar radiation and other meteorological factors (e.g., temperature, pressure, humidity). A comparative analysis against other machine learning techniques such as linear regression, recurrent neural networks, Adaboost, and support vector machine is presented. The most common metrics, such as R${}^{2}$, MSE, and RMSE, are used to evaluate the performance. An advantage of this improvement is that it is universal for RF, as it can be adjusted to all zones and various combinations of additional climatic variables. Another advantage is the high level of prediction without adjusting or modifying the data obtained. Despite being a tree-based algorithm, it does not have too much overfitting.

### Study Area

Mexico alone uses one-fifth of all the energy produced in Latin America, and demand is growing. Most of this energy is produced through natural gas and oil; this is a significant concern, as in order to achieve the proposed goals it is necessary to reduce the production of pollutant gases by 40 percent by 2030 [12,13].

One of the ways to achieve this is through the implementation of renewable energies; as development in this area has been slow, Mexico has a great potential for innovation and development. By 2030, solar photovoltaic energy will account for about 30 GW of energy annually through hybrid schemes of distributed PV power plants and mini-grid applications.

The way to measure the solar energy potential of a territory is through measuring the solar radiation. According to [12,14] Mexico is located between 15° and 35° north latitude, as can be seen in Figure 1, belonging to the most favored region for solar resources. It receives a daily average of 5.5 to 6.3 $\frac{\mathrm{kWh}}{m^{2}}$ and a solar energy generation of 114.2 GWh [14]. This represents 60% more than the real potential of countries such as Germany [12]. The northwestern part of Mexico has excellent potential for solar PV power generation. Average daily irradiation in the region can exceed 8 $\frac{\mathrm{kWh}}{m^{2}}$ in spring and summer [13].

According to the [15], only systems that convert 15% of the electrical energy, such as a 25 km${}^{2}$ PV plant in the state of Chihuahua or the Sonora desert (0.03% of Mexico), could supply Mexico’s electricity needs. The typical direct isolation of northern Mexico is equivalent to the best deserts in North Africa and the southwestern United States, where many utility-scale solar power plants are being built today. As shown in Figure 1 the central region of the midwest has abundant solar resources, as does the Baja California peninsula.

**Figure 1 micromachines-13-01406-f001:**
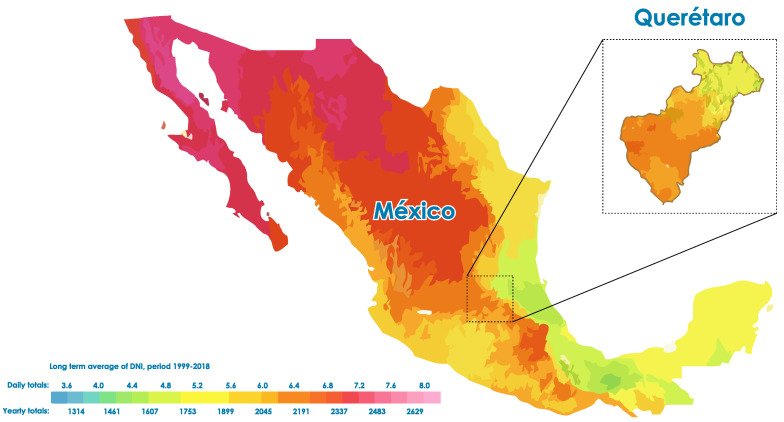
Solar atlas map of Mexico [16].

For this work, we chose the city of Queretaro, the capital of the state of the same name. It is located in the geographical center of Mexico, at 20.612137${}^{{}^{\circ}}$ N, 100.410217${}^{{}^{\circ}}$ W, with an average altitude of 1820 m above sea level. In the rainy season, the city is very cloudy, while in the dry season it is partly cloudy, and it is hot all year round.

In [17], the author details Queretaro as having the potential to produce an additional 1178 $\frac{\mathrm{GW}}{h}$, which places it in fifth place in the Bajío region and above countries such as Germany. However, it currently only produces 8 $\frac{\mathrm{GW}}{h}$ through biomass, or just 0.6% of its potential [12]. The clean energy source with the most significant potential for the state of Queretaro is geothermal energy, which can produce up to 649 $\frac{\mathrm{GW}}{h}$. Next in potential are wind and solar energy, possibly producing 250 $\frac{\mathrm{GW}}{h}$ and 191 $\frac{\mathrm{GW}}{h}$, respectively [18]. Recent studies have calculated that the capacity for the city to generate energy through sunset is between 6 KWh and 7 KWh [19].

## 2. Related Works

This section presents a compilation of the different types of machine learning algorithms used to predict the solar radiation variable.

In order to carry out this review of the state of the art, academic search engines such as Scopus, IEEE, Semantic Scholar, and MDPI were consulted. In addition, other innovative search engines such as Google Scholar, Microsoft Academic, and Connected Papers were used on account of their significant power and robustness. The use of Mexican institutional repositories such as those of the Autonomous University of Queretaro (UAQ) and the National Polytechnic Institute (IPN) were used; based on their importance, an analysis of scientific journal articles, conference readings, books, and theses of all degrees (PhD and Masters) was carried out. The works were filtered according to the following keywords:Machine learning (RF, SVM, ANN, Kernel regression) algorithms for solar radiaton predictionHybrid algorithms (combination of many algorithms) for solar radiaton prediction

Table 1 below contains the results of the search; the following representative characteristics were broken down:The performance metrics used by the authors identified the most used approaches, namely, R2, MAPE, MSE, and the accurac of models based on these approaches. Other performance metrics, such as the MAE or MBE, were reported as well, although only in three articles.Models that only predicted one type of variable (SR vs. SR and wind) have a better outcome than those that try to predict two or more variables.The most used ML types were identified, as were the extra algorithms used to improve their performance, with heuristic and evolutionary algorithms being the most common.The type of solar radiation most studied in these works was global radiation, with direct or diffuse radiation mentioned in only two articles.In order to determine which types of hyperparameters were used in each model, the most representative alternatives for optimizing the RF algorithm were considered. In Section 3.5, we provide a list of the hyperparameters we used and how each affected performance in this study.

Pang et al.  [20] used the recurrent network algorithm (RNN). The authors used Alabama weather data to feed the model and predict solar radiation. A comparison between RNN and MLP was realized. They used input variables such as radiation and temperature and calculated others such as the hours of the day. They obtained an R${}^{2}$ of 0.983 and an RMSE of 41.2 against an R${}^{2}$ of 0.974 and an RMSE of 55.7 for the ANN. They proposed using variables that affect prediction, such as cloudiness, and their application to control models

Zhu et al. [21] compared an LSTM algorithm and a convolutional neural network (CNN). Unlike other research works, they used images instead of historical data. They achieved a 0.93% prediction efficiency and an RMSE of 29.92. The advantage of using images is that they obtain a good result even in very cloudy or rainy conditions.

Shamshirband et al. [22] performed a comparative analysis of the different algorithms that make use of ANNs. They were able to find the following disadvantages: MLP-based networks cannot capture long-term memory due to their architecture, as the hidden layers are not connected, making their performance poor when working with time series. LSTM networks improve upon this disadvantage, and are thus used in prediction work thanks to their robustness.

Tree-based methods are very popular because of their easy implementation and because it is possible to optimize prediction by adjusting the hyperparameters. Meng et al. [23] used an RF algorithm to perform RS predictions in Xingtai City, northern China; they used three datasets, one with sunny days, one with cloudy days, and a third with rainy days. The regression results on each data set were averaged to obtain the final results. They obtained an accuracy of 95% for sunny days, 90% for cloudy days, and 82% for rainy or snowy days.

Lee et al. [24] presented an investigation comparing different machine learning tree models (RF, Bagged Tree, and Boosted Tree), reporting an R${}^{2}$ of 97% with an RMSE of 59.27 compared to an SVM algorithm and Gaussian regression. They used data from sixteen different stations. They noted that their results do not show a preferred model that stood out from the others, while concluding that the tree-based models perform well.

Hybrid algorithms, which are the combination of two or more different algorithms, are not limited to machine learning alone, and can combine traditional statistical algorithms such as linear regression and Pearson correlation models with metaheuristic or optimization algorithms such as Grey Wolf or Monte Carlo optimization.

Eseye et al. in [25] proposed a novel model based on Wavelet-PSO-SVM, obtaining data from meteorological sources and statistical models. To perform their calculation, they divided their dataset for the four seasons of the year and obtained the following results: in winter, an RMSE of 0.73 for a horizon of 3 hours; in spring, an RMSE of 0.76; in summer, 1.024; and in winter, 0.8598. These results demonstrate the effectiveness of this approach for short-range prediction models.

**Table 1 micromachines-13-01406-t001:** Research papers with metrics used with most common machine learning algorithms.

Year	Reference	Classification	Algorithm	Kernel (only for SVM)	R${}^{2}$	Mean Absolute Error (MAE)	Root Mean Square Error (RMSE)
2017	Quej et al. [26]	Machine Learning	SVM/ANFIS/ ANN	RBF	0.689	1.973	2.678
2018	Gupta et al. [27]	Hybrid	RF + Genetic	–	24.45	–	47.74
2019	Ghazvinian et al. [28]	Hybrid	SVM + PSO	RBF	9.01–0.141	–	19.10
2019	Srivastava et al. [29]	Machine Learning	Random Forest	–	–	15.74	65.08
2020	Sun et al. [30]	Tree Based	Adaboost	–	–	8.41	–
2020	Pang et al. [20]	Neural Network	RNN/MLP	–	0.983	–	41.2
2021	Zhu et al. [21]	Neural Network	CNN, LSTM	30	0.958		23.89
2021	Aljanad et al. [31]	Hybrid	BPNN-PSO	–	0.7537	–	1.7078
2021	Philibus et al.[32]	Hybrid	ANN/SVM	RBF	0.9112	0.1842	0.1014
2022	Faisal et al. [33]	Neural Network	RNN, LSTM, GRU	–	128	0.918	0.958
2022	Brahma et al. [34]	Deep Learning	Multi-Step CNN Stacked LSTM	–	68.62	9.721	9.859

## 3. Theoretical Bases

This section presents the methods used to determine the correlations between the weather variables in a preliminary dataset analysis and presents several of the essential features of Random Forest (RF), including optimization of the hyperparameters, the algorithm, and the most common metrics used to check the performance of the machine learning algorithms used in this article.

### 3.1. The 80–20 Rule

The Pareto principle, sometimes called the 80–20 rule, is a fundamental principle of distribution and is based on the principle that, statistically speaking, many activities in the world can be explained in this way. There are other ways to partition training data, such as the training–validation–test ratio of 80–10–10. There are no studies where either is required as a general rule. In this research, we consider the method of scaling the data for the first iteration of the dataset with 38,000 solar radiation data points measured per minute. In order to simplify the results, a new dataset was reformulated, with the maximum values for each hour considered. The results obtained were very similar for both cases, and in order to simplify the process, remembering that it may be used with a low resource system, the 80–20 split was considered [35].

### 3.2. Pearson Correlation

Correlation is a measure taken from statistics that serves to measure or calculate how close two variables are to each other. The most popular method is the Pearson product–moment correlation, which allows the linear relationship between two variables to be observed. Many works have indicated that a Pearson value equal to zero does not imply that the variables are independent of each other [36,37].

### 3.3. Distance Correlation

A recent significant contribution to the field of statistics is the distance correlation, which serves as a measure of dependency between two paired random vectors that do not necessarily have to be of the same dimension. It was introduced by Szekely et al. [38] to deal with a deficiency in the Pearson correlation involving dependent variables that can have a value of zero, which can lead to the mistaken conclusion that the relationship between two variables are uncorrelated when in fact the correlation is merely a nonlinear one.

Distance correlation addresses this issue by measuring the strength of the association between nonlinear random variables. The distance correlation is from 0 to 1, where 0 implies independence between X and Y and one implies that the linear subspaces of X and Y are equal. This extends Pearson’s correlation because it can detect nonlinear associations and works in a multidimensional way.

### 3.4. Random Forest for Regression

RF is one of the most widely used machine learning algorithms due to its simplicity. It can be used in the areas of regression and classification. It is one of the class of supervised learning algorithms and SVMs, which includes naive Bayes algorithm and other tree-based algorithms such as Adaboost. It was first developed and proposed by Breiman et al. [39] at the University of California in 2001.

RF consists of combined decision trees that generate a more accurate and stable prediction; in general, the more trees in the ensemble, the more robust the results. This model increases the randomness of the decision tree model and grows the trees; instead of splitting a node, it searches for the best feature among a random subset of features.

The important feature used in this project is the out-of-bag error or OOBE, also known as the generalization error, which is a kind of cross-validation already included in the algorithm, namely, the average prediction error of the first observations. The OOBE is used to estimate the generalization capability of the algorithm [23]:(1)$$\mathrm{OOBE}=\frac{1}{N}\sum_{i=1}^{N}{\left(Y_{i}-\hat{Y}\right)}^{2}$$

The last characteristic is the variable importance measure (VI), which is obtained by permuting a feature and averaging the difference using the OOBE before and after the permutation over all the trees [40]:(2)$$\mathrm{VI}(X^{j})=\frac{1}{q}\sum_{l=1}^{q}\left(\bar{OOBE_{l}}-OOBE_{l}\right)$$
where $\bar{OOBE}$ is the average of the estimated OOBE. Figure 2 presents the general structure of the operation of the RF algorithm.

### 3.5. Hyperparameter Optimization

Hyperparameters are the settings that the user can arbitrarily configure before starting the training process to optimize the performance of the model, e.g., in the case of tree-based algorithms, the number of estimators and the number of decision trees. In contrast, model parameters, such as weights in the case of neural networks, are learned during the model training process. Hyperparameters are already available by default for most existing programs or libraries. However, these are not always the most appropriate. It is impossible for the user to determine which are optimal ahead of time, making the science of machine learning a trial-and-error endeavor.

In the beginning, the tuning of these hyperparameters was more dependent on experimental results than on theory. Over time, various combinations of hyperparameters were tested until the best result was achieved, which led to the fundamental problem of overfitting. If a model is very well optimized for a particular data type, it becomes be challenging to fit varied or new data types. One way to avoid this is through the cross-validation of results.

When searching for a universal algorithm to apply to new problems, the hyperparameter fitting problem can be seen as an optimization problem in which optimizing model performance involves optimizing the objective function. There are different algorithms for hyperparameter optimization.

### 3.6. Finding the Best Combination of Tunable Hyperparameters for the Random Forest Regressor

In a Random Forest approach, there are three main parameters to adjust:**n_estimators** In general, more estimators is better, although it has the disadvantage of decreasing yield. As is evident, more trees leads to a longer computation time. One of the great challenges of this algorithm is to find the critical number of decision trees where the best accuracy is obtained while balancing the computation time.**max_features** This is the maximum number before cutting to a new node. If certain trees consider a different subset of features than others, the correlation between those two groups should be minimal. This is desirable because it allows the influence of each feature to be assessed individually.**maximum_depth** Having trees with too much depth effectively leads to overfitting. There is a critical depth at which trees are split deep enough to obtain a useful fit without being overly influenced by individual values. A depth constraint can be created by adjusting the parameters (min_samples_split), (min_samples_leaf), (min_weight_fraction_leaf), or (max_leaf_nodes), rather than specifying a preset value for the depth.

To find the best hyperparameters, a programming function named RF optimizer Algorithm 1 is presented; the inputs are n = the number of estimators, f = the number of features, and s = the number of samples, which are then proceed to find the best score using the fitness function of the model.

In Figure 3, the flowchart shows the way in which the program works to extract the data from the dataset and divide it into test, training and validations sets, after which the machine learning algorithms are applied, including hyperparameter optimization for random forest and Adaboost and metrics validation.
**Algorithm 1** Stages of the proposed Random Forest Optimizer.  **Input:** estimators, features, samples.  **Output:** model, bestE, bestF, bestS.1:**Function** RF Optimizer(*x*, *y*)2:best_score=float                                                                             ▹ initialize score3:**for all** *n* **do**4:      **for all** *f* **do**5:          **for all** *s* **do**6:               $model=RF(oob\_score=True,estimators=n,features=f,samples=s,n_{j}obs=-1)$7:               model.fit(x,y)                                                                                   ▹ (b), *n*8:               **if** model.oob_score≥best_score: **then**9:                   $best\_score,bestModel,bestE,bestF,bestS\leftarrow model.oob_{s}core_{,}model,e,f,s$10:return model,bestE,bestF,bestS                                               ▹ calculated values

### 3.7. Statistical Metrics for Data Validation

In this subsection, we present metrics and indicators for evaluating prediction models; these were selected from the state-of-the-art review in Section 3. Then, we determine the accuracy of the data. Although the most reliable metrics for predictions are MAPE and accuracy, we include four of the most popular metrics obtained from the review of previous works in [32].

The mean square error (MSE) is perhaps the most straightforward function that can be calculated in machine learning. It takes the difference between the model predictions and the actual data or ground truth, squares it, and applies the average to the entire dataset. MSE can never be negative:(3)$$\mathrm{MSE}=\frac{1}{N}\sum_{i=1}^{N}{\left(y_{i}-\hat{y}\right)}^{2}$$

The root mean square error (RMSE) calculates the goodness of fit, which is related to preventing very high errors:(4)$$\mathrm{RMSE}=\sqrt{M}SE=\sqrt{\frac{1}{N}\sum_{i=1}^{N}{\left(y_{i}-\hat{y}\right)}^{2}}\;$$

The mean absolute percentage error (MAPE) serves the function of calculating the absolute error in percentage of predicted or observed variables [41]:(5)$$\mathrm{MAPE}=\frac{1}{N}\sum_{t=1}^{N}\;\left|\frac{y_{i}-\hat{y}}{y_{i}}\right|*100$$

The mean absolute error (MAE) is an indicator of the performance of the prediction model, achieved by observing how close the predicted variables are to the observed variables:(6)$$\mathrm{MAE}=\frac{1}{N}\sum_{i=1}^{N}\left|y_{i}-\hat{y}\right|$$

R-squared, or the coefficient of determination, is the statistical measure that measures in percentage how close the data are to the regression line [42]:(7)$$R^{2}=1-\frac{\sum {\left(y_{i}-\hat{y}\right)}^{2}}{\sum {\left(y_{i}-\bar{y}\right)}^{2}}$$

These indicators provide insight into the effectiveness of machine learning models if the value of the statistical indicator is zero, as in the ideal case.

## 4. Materials and Methods

This section explores the study area and how the data were acquired, including the climatological characteristics of the area, the tools used to process the data, and the requirements when working with the data to perform predictive analysis.

### Dataset

The data were collected and compiled from the University Network of Atmospheric Observatories (RUOA), campus Juriquilla, Queretaro, which belongs to the National Autonomous University of Mexico. Its policy explains that it is open data for public access. There is only one observatory in the city, located at the following coordinates: 20.7030${}^{{}^{\circ}}$ N, 100.4473${}^{{}^{\circ}}$ W, altitude 1945 m.a.s.l. The process of recording data is as follows: the data acquirer is programmed to record data every minute, and after one hour it delivers an average. The variables recorded are wind speed, wind direction, air temperature, atmospheric pressure, rain, relative humidity, and SR. These date can be found on the following website: [43]. The captured data used here were from the years 2020 and 2021, which allows a comparison analysis to be carried out in order to check the performance of the algorithm.

The dataset consists of 52,698 samples, with the maximum of each value in each hour as follows:SR: $\frac{W}{m^{2}}$Temperature: ${}^{{}^{\circ}}C$Humidity: measured in percentBarometric pressure: HgWind direction: measured in degreesWind speed: $\frac{\mathrm{km}}{h}$Sunrise/sunset: Queretaro Time (GMT-5)

After adding the UNIXTime column, it was necessary to transform the date in DD/MM/YYYY format to MM/DD/YYYY and then convert it to UnixTime with a batch tool.

This research only takes sunshine hours into account in order to filter out the nighttime radiation data captured by the weather sensors of the station, which do not have a significant impact on the prediction.

By capturing the sunrise and sunset times in the database, it is easy to provide the algorithm with this functionality through Equation (Equation 8):(8)DayLength=Sunset−Sunrise

Figure 4 below represents the methodology followed to perform solar radiation prediction with different algorithms. All of these steps are broken down and reflected in the results in Section 5.

## 5. Results

This section presents the results obtained from this research and follows the methodology presented in Figure 4.

Nevertheless, before performing the analysis it is necessary to prepare the data, maintain a high-level structure to use with any method, and ensure that our results have the same integrity. We used a previously described method for this, with the following steps:Remove corrupt or unreadable data;Replace all non-numerical data in all columns with numerical and floating values;Replace all missing or zero values through normalization techniques;Apply label coding to all columns.

To determine the subset, we start by splitting the original set into training and test sets in the ratio 80–20%. This subset consists of samples of all features with a fixed random generator to provide homogeneous results across all ML models, transforming an unbalanced training dataset into balanced data subsets.

### 5.1. Heat Map and Pearson Correlation

After the reading and adequate data transformation are complete, the next step is to visualize them. As a first step, a correlation matrix is constructed to identify the relationships between the most significant variables. This can be observe in Figure 5.

As it appears in Figure 5, with a score of 0.56, the Pearson correlation matrix shows that there might be a relationship between solar radiation and temperature, and to a less extent with humidity, with a score of −0.47. Values between +1 and −1 (for example, r = 0.8 or −0.4) indicate that there is variation around the line of best fit. Pressure, with a score of −0.04, indicates a slight relationship, although pressure is strongly related to temperature at −0.55 and humidity at 0.47; this behavior can be seen in Figure 6.

### 5.2. Distance Correlation

Because the Pearson coefficent is not sufficient to state that there is a relationship between two variables, additional metrics such as the correlation distance were used. The following Table 2 shows the obtained values of the correlation distance for the Queretaro 2021 dataset.

The following conclusions can be drawn from this data exploration:The higher the detected temperature, the higher the amount of SR; this can be seen from the Pearson correlation value *R* of 0.56 and the relationship observable in Figure 7a between radiation and temperature on hourly and weekly scales. The distance correlation of 0.55 means that there is a relationship between temperature and SR.Humidity has an inverse or negative relationship with RH compared to temperature, and is potentially significant and cannot be ignored as a potential driver in the climate system. This can be observed in Figure 7 on both the hourly time scale and to a lesser extent on the weekly scale.Based on both the Pearson and distance correlations, pressure does not seem to have a direct relationship with SR, although it is related to temperature and humidity. Related work can be found in [44].The variables of wind speed and direction variables are not relevant for this study; while a correlation can be observed, this does not mean imply any causality.While Queretaro is a city without significant seasonal changes, during the rainy and summer period significant changes can be observed, as can be seen in the weekly graphs in Figure 7.The weekly time scale is the best for forecasting. The month-to-month variations are substantial and do not capture the seasonal changes over the course of the year. Minute time scales are helpful for more refined work such as PV control systems. Hourly time scales are good, although they can be noisy if not filtered properly.

### 5.3. Machine Learning for Solar Radiation Prediction

It is desirable to obtain an optimized RF-based algorithm that is able to predict values of SR for a number of inputs. As mentioned in Section 2, there are several different models to choose from, and more than one may be appropriate. From the previous analysis involving testing of the different models and evaluation of their performance in predicting SR, the most appropriate models are:Recurrent NN regressionSVM regressionRandom Forest regressionAdaboost regressionOptimized RF regressionOptimized Adaboost regression

### 5.4. Simple Linear Regression

Linear regression is one of the most widely used methods in data science for establishing relationships between dependent and independent variables employing a straight line. As such, it serves as a benchmark to compare performance against more advanced ML models. The accuracy obtained in this research was 55.23%, and the R${}^{2}$ score was 0.55. Figure 8 shows the regression plot and the graph from the predicted versus the real data.

### 5.5. Recurrent Neural Network Regression

Neural networks for prediction can be adapted to many problems; in this case, RNN with default values was used, although we were aware that this method may not be the most suitable for the proposed model. Figure 9a displays the regression of the model.

From Figure 9a,b it is possible to see the poor results obtained using standard parameters. Again, RNN could not properly train on the data, and every point was moved to the zero position. It would be possible to obtain better results with this algorithm if different input values for biases, numbers of neurons, or activation functions were provided.

### 5.6. Support Vector Regression

The prediction results obtained by SVM were inferior, with an accuracy score of −51% and an R${}^{2}$ of −0.51 for the 2021 dataset. According to the technical documentation, a negative R${}^{2}$ result means that the model had terrible training. Because R${}^{2}$ compares the goodness of fit of the model (SVM) with that of the null hypothesis in the regression (a horizontal line), the experiment was repeated throughout with the other two datasets, obtaining similar performance (see Tables 5 and 6). The run time for this algorithm was extremely long.

Regression and comparison plots were performed in order to compare the methods. Figure 10a shows that the data could not be adjusted to the horizontal line. Figure 10b again shows the null results obtained by the base SVM algorithm. Re-testing the SVM with other kernels, SVR (kernel = “linear”, C = 100, gamma = “auto”) or SVR (kernel = “poly”, C = 100, gamma = “auto”, degree = 3, epsilon = 0.1, coef0 = 1), achieved similarly poor performance. While further optimization of SVM was beyond the scope of this paper, interested readers can find further studies involving SVM for solar radiation prediction and forecasting in [45,46].

### 5.7. Random Forest Regression

To perform this regression, only the default hyperparameters provided by the library (n_estimators = 100, max_samples = None) were used as a basis for comparison against the optimized RF algorithm.

The data obtained using RF as a regression model had a 90.34% accuracy score and an R${}^{2}$ of 0.907. Figure 11a evinces improved training data, while Figure 11b exhibits an improved fit between the test data and the predicted data. While there are areas where the algorithm fails, the accuracy is much improved overall.

### 5.8. Adaboost Regression

Adaboost is another tree-based machine learning algorithm; it achieved an efficiency score of 89.13 %, although it seems from Figure 12a that the regression of the training results vs. the actual data does not work very well. This could explain the large MAPE number as well. In Figure 12b, it is noticeable that the predicted data are well adjusted to the real data.

### 5.9. Optimized Hyperparameters Random Forest

Using the proposed improvements presented in Section 3.6 and applying the algorithm proposed in Section 1, the following results were obtained after determining the best parameter values as seen in Table 3. In Figure 13b, a slight improvement can be observed, with 94.68% in accuracy and a R${}^{2}$ of 0.94, which is a small but significant improvement against 90.34% achieved with the traditional RF, which is already quite good.

### 5.10. Numerical Results

In order to obtain accurate results, it is necessary to present the values of the metrics and indicators. The pseudocode used to calculate the regression metrics proposed in the “metrics” subsection is provided below. It is possible to estimate each parameter using prediction and training as inputs.

Table 4 contains the values of the calculated metrics for all the studied models using the 2020 dataset. The optimized RF model has the best performance at 94%, followed by the non-optimized RF model at 90%; in last place is SVM with −54% accuracy. Other studies, such as [29], obtained similar results when implementing RF for SR prediction. The values obtained here validate the regression plots, as in the case of LR and MPL; the values tend to zero not because of null values, but rather as a reflection of the difficulty these models had in providing good predictions.

As discussed in the methodology section, it is necessary to validate these results in different scenarios. Table 5 shows the results calculated for the same area in 2021. This dataset, although incomplete, led to similar results as those found with the previous dataset, with the proposed optimized RF having roughly 95% accuracy and the other models continuing to behave the same.

Finally, a standard dataset with data from Hawaii for the year 2016 was used as the basis for Table 6 where, again, very similar results to the previous ones are obtained and RF obtained a 93% accuracy. Checking the results in this way at two different sites and times further validates the results presented in this article.

## 6. Discussion

This section presents a discussion of the results obtained in comparison to the results obtained by other authors.

Although multiple previous works are related to improving the performance of prediction models, this work is distinguished by presenting two main contributions. The first contribution is that the proposed model considers the sunshine hours variable, which affects the quality of the solar radiation measured in certain zones. This variable can be calculated using public data from automated weather stations. The second contribution is the presentation in this work of an optimized RF algorithm with the best hyperparameters obtained by Algorithm 1. This contribution can be compared and contrasted with the results of previous works using Table 7.

Of the sources mentioned above, the algorithm presented by Chaibi et al. obtained 94% accuracy using Bayesian optimization [47]. Zou et al. used a radically different approach involving an LSTM algorithm and meteorological variables such as relative humidity and Aerosol Optical Depth instead of daylight hours, obtaining 90% prediction accuracy [48]. Sreekumar et al., considered cloud movement and cover for their prediction, achieving 97.11% accuracy [49]. Rahman et al. [50] achieved a respectable 98% accuracy using the same variables as ours except for sunshine hours with a simple neural network and a modified ADAM optimizer. The best scores we found were reported by Fuselero et al. [51], who obtained 98% prediction accuracy using the same input variables. However, our work nonetheless presents excellent performance using the proposed optimization method, reporting an accuracy of 95.98%.

Different machine learning algorithms were used in this study and obtained different results despite using the same dataset. This may be due to several factors.

Regarding the distribution of data, for example, SVM-based algorithms tend to have improved performance on small datasets; on the contrary, as can be seen in our results, tree-based algorithms (RF, Adaboost, XGboost) tend to achieve excellent results on larger datasets.In the preliminary inspection of the data, we found several instances in which no data was collected, especially in December and early January, presumably due to a lack of personnel to maintain the site. Much time was consumed ensuring that the dataset was complete and that the information was accurate and reliable, beyond the preparation described in this article.Another reason is that the parameter selection of the machine learning models could not find the optimal global solution.Performing a statistical analysis of the data is essential, especially for time series, including finding the mean, maxima, and minima of the data, checking its variance, and determining whether the type of regression or prediction method is appropriate.Several articles have pointed out that different meteorological factors such as air temperature, relative humidity, wind speed, and precipitation are closely related to solar radiation, although the effects and the degrees to which they affect it can vary according to the different regions or countries under investigation.Differences in the implementation of algorithms can directly affect both performance and the obtained results. There are many ML libraries available for research such as that described in this article, including libraries such as Pytorch, scikit-learn, and SciPy, among others.

## 7. Conclusions

In this study, historical meteorological data were collected for the last two years for the city of Queretaro in the Juriquilla area, both daily and with an interval of one hour. Models were chosen according to the collected literature, and regression and prediction studies of the models were performed. The ability of the RF algorithm to use historical feature values allows it to outperform other deep learning models in time series applications. It achieved up to 95% prediction accuracy. An essential limitation at the time of the study was the lack of reliable data. Although there are different stations around the city, none have standards for data collection, making it challenging to create datasets and achieve consistent results. We determined that the climate in this area is very favorable for such a study, as it is very consistent most of the time, which helped to create a very accurate model.

Based on our experiments, it can be concluded that:There is a relationship between the variables of SR and temperature, as proven by simple linear regression; this relationship is dependent on the humidity in the city.The simple linear regression model has a good fit at the time of prediction.The random forest regression model has the best fit, as at the beginning of SR prediction the proposed optimization only managed to obtain a 3–4% increase (depending on the used dataset).Our results can serve as a basis for future experiments, work with other types of neural networks such as recurring ones, application of kernel regression, etc.The data obtained in this study can be used in solar control systems, such as PV MPPT systems, to optimize their efficiency.Our results can optimize the placement of renewable energy plants throughout the state of Queretaro.

### Future Work

Future work will involve an exhaustive statistical analysis of time series to determine whether tree-based algorithms are best for this application. Such work could focus on expanding the scope of the present study to other areas, such as wind and humidity studies.

With the data obtained here, it may be possible to generate accurate forecasting and nowcasting models of climatological variables, specifically, solar radiation.

Use of other ANN models, such as recurrent neural networks (RNN), could be investigated, as these achieve excellent results on time series and nonlinear data.

Generating an empirical prediction model and taking into account characteristics (temperature, pressure, humidity) that are not entirely independent by incorporating their mutual relationships and influences into the algorithm could create a more realistic model.

Applying these models to weather station websites would allow us to determine which algorithms perform better in real time.

A more accurate RF model could represent an alternative if more emphasis were placed on the following areas.

Tree-based models allow for extracting the importance of different characteristics in determining the regression model. By looking at the importance of the different features, it is possible to understand whether the two models assign the same importance to different parameters.The RF algorithm performed very well from the beginning, without any need to optimize hyperparameters or specific data; yet, this selection was purely arbitrary, without any way to check whether they were the most suitable for the task.Iterating through a list of arbitrary values and choosing the one that provides the best results is not the best approach to optimization. It would be beneficial to spend more time tuning, perhaps including an analytical method for determining optimal values.Other features, such as ozone and contaminant gases, could be considered, as these features are known to impact light transmission through the atmosphere, especially at certain wavelengths.

From the data, it appears that the following drawbacks characterize RF regression:It is characterized by a high variance in predictionIt appears to systematically overestimate SR after sunset

Neural networks used for prediction can be adapted to many problems; in this case, we used multiple perceptron regression with its default values, even knowing that these may not be the most suitable for this model.

## Figures and Tables

**Figure 2 micromachines-13-01406-f002:**
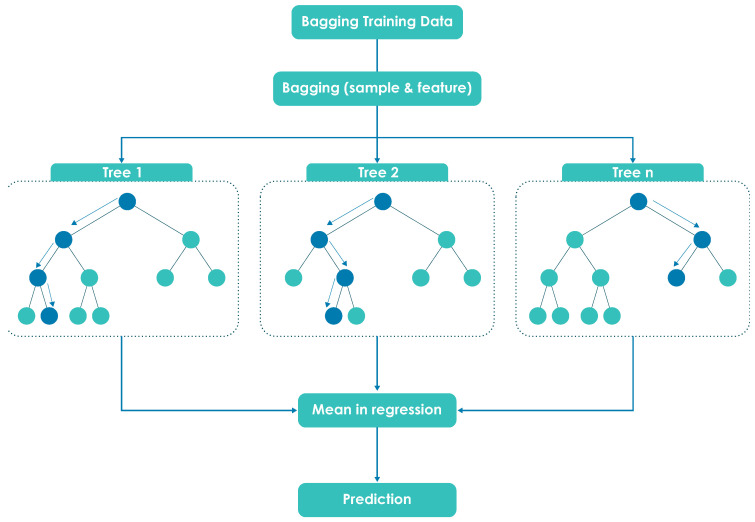
Random Forest regression tree.

**Figure 3 micromachines-13-01406-f003:**
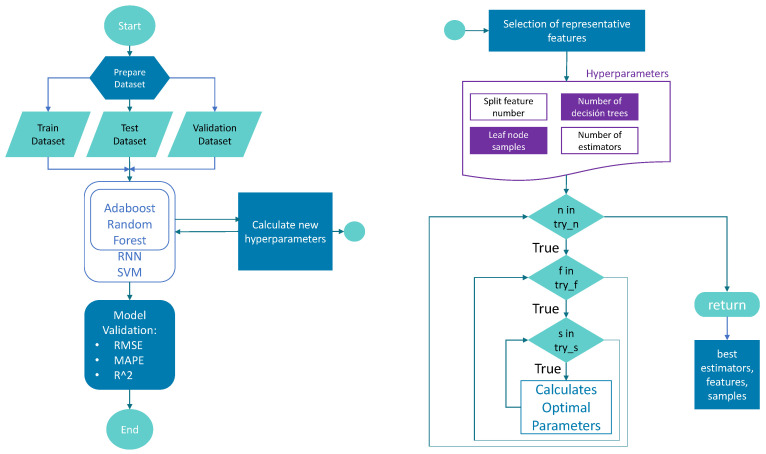
Flowchart with the proposed improvements applied to random forest regression algorithms.

**Figure 4 micromachines-13-01406-f004:**
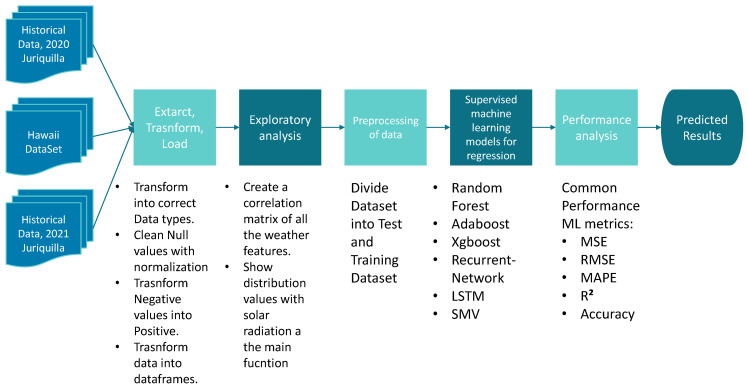
Graphical representation of the methodology.

**Figure 5 micromachines-13-01406-f005:**
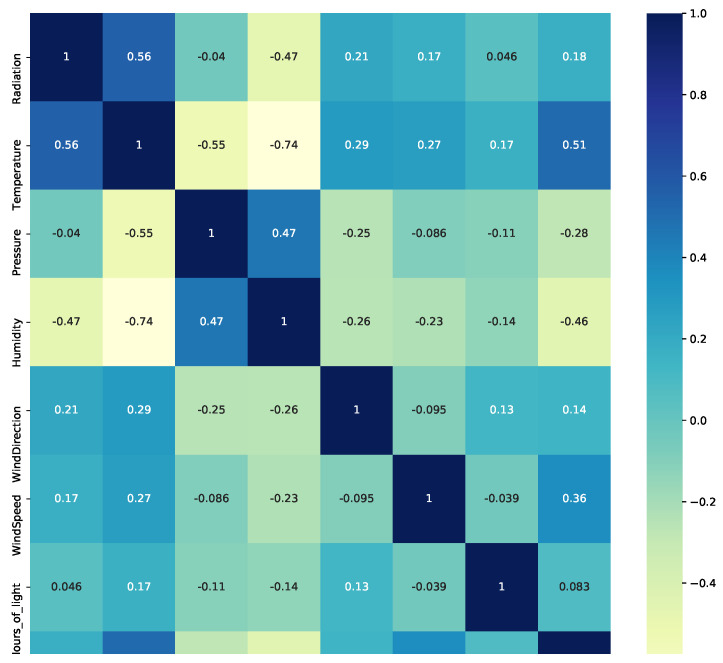
Correlation matrix plot for the all the weather variables.

**Figure 6 micromachines-13-01406-f006:**
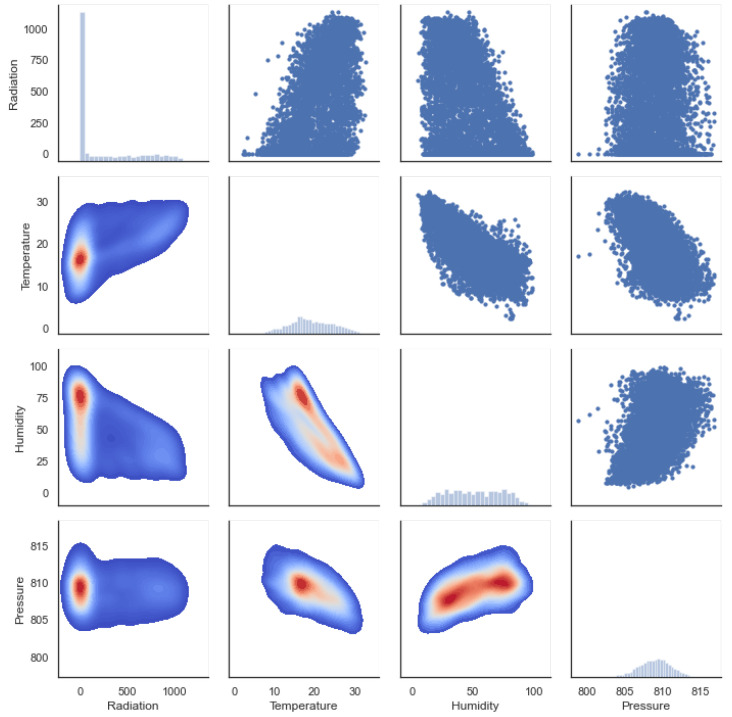
Scatter plots of SR as a function of various features.

**Figure 7 micromachines-13-01406-f007:**
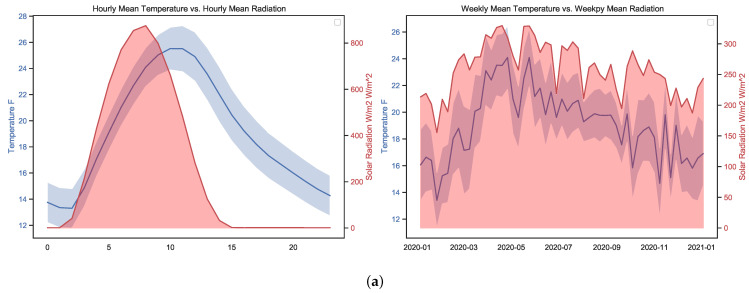
Hourly and Weekly mean data with SL as the principal feature.

**Figure 8 micromachines-13-01406-f008:**
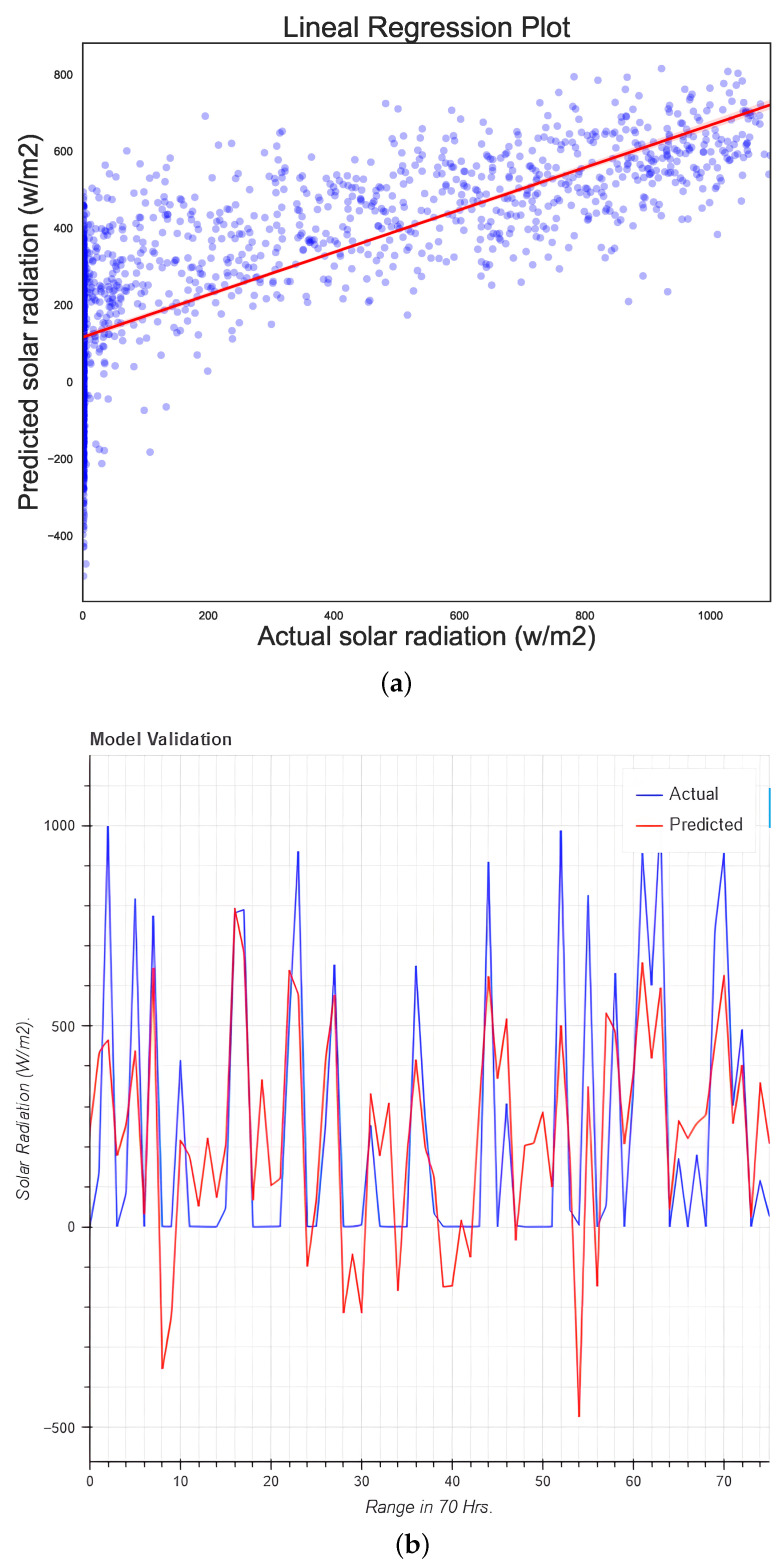
Regression plots for (**a**) the linear regression model and (**b**) the test data.

**Figure 9 micromachines-13-01406-f009:**
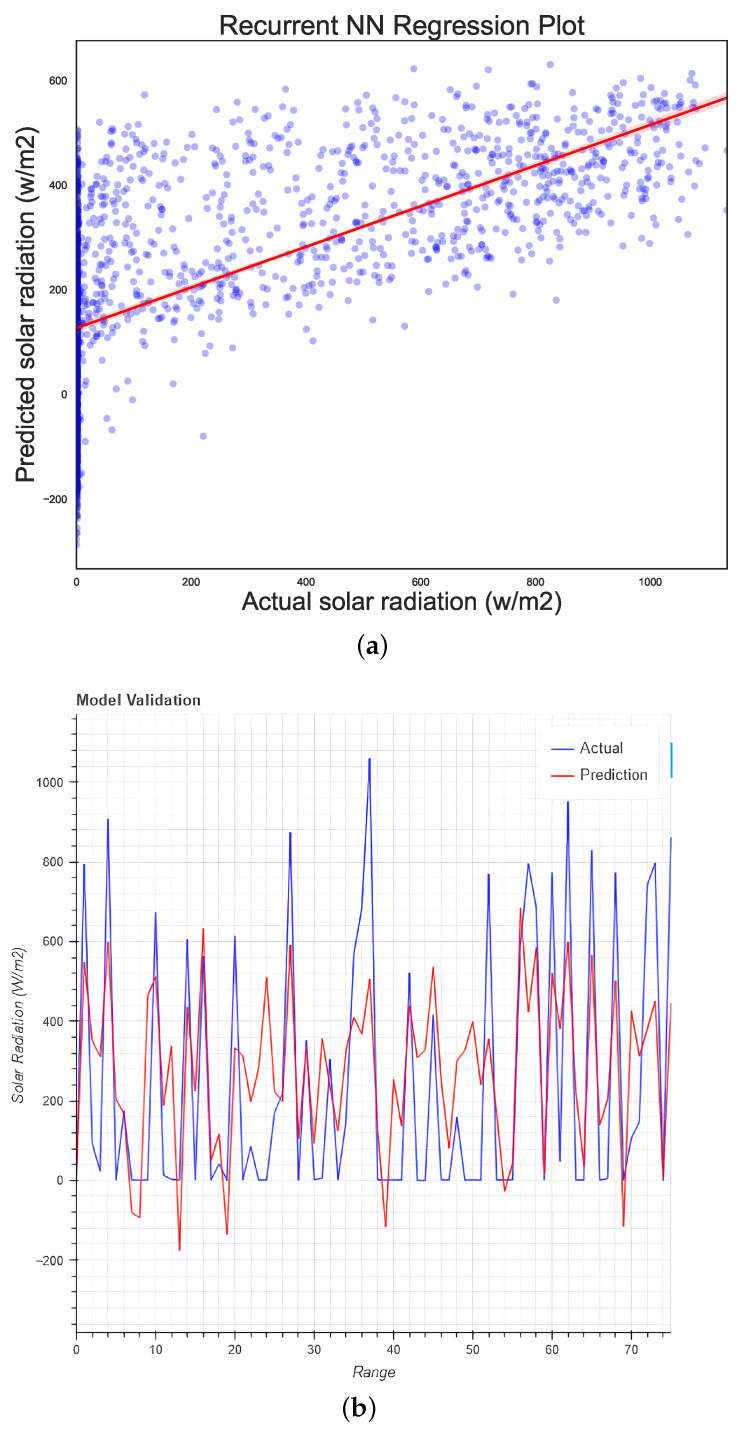
Regression plots for (**a**) the RNN model and (**b**) the test data using RNN regression.

**Figure 10 micromachines-13-01406-f010:**
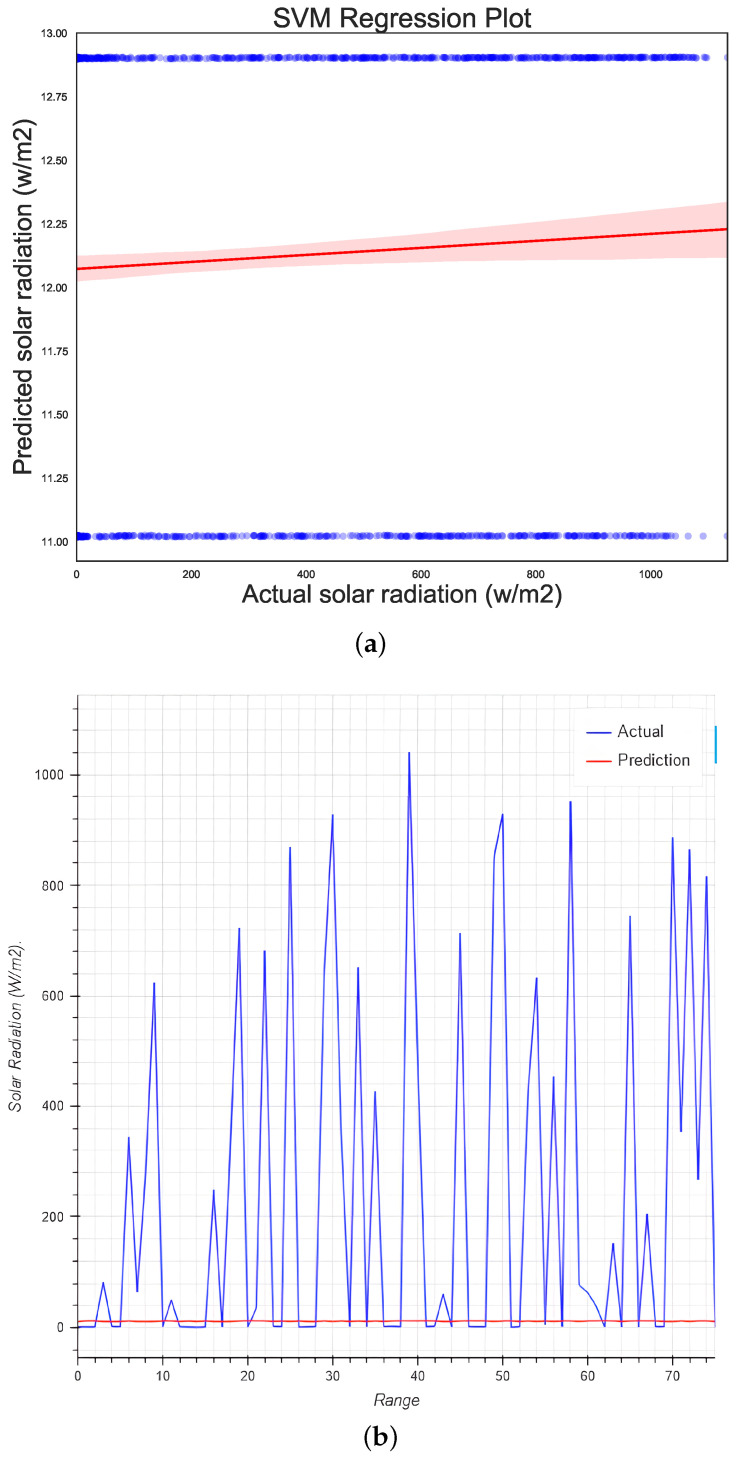
Regression plots for (**a**) the MPL model and (**b**) the test data using MPL regression.

**Figure 11 micromachines-13-01406-f011:**
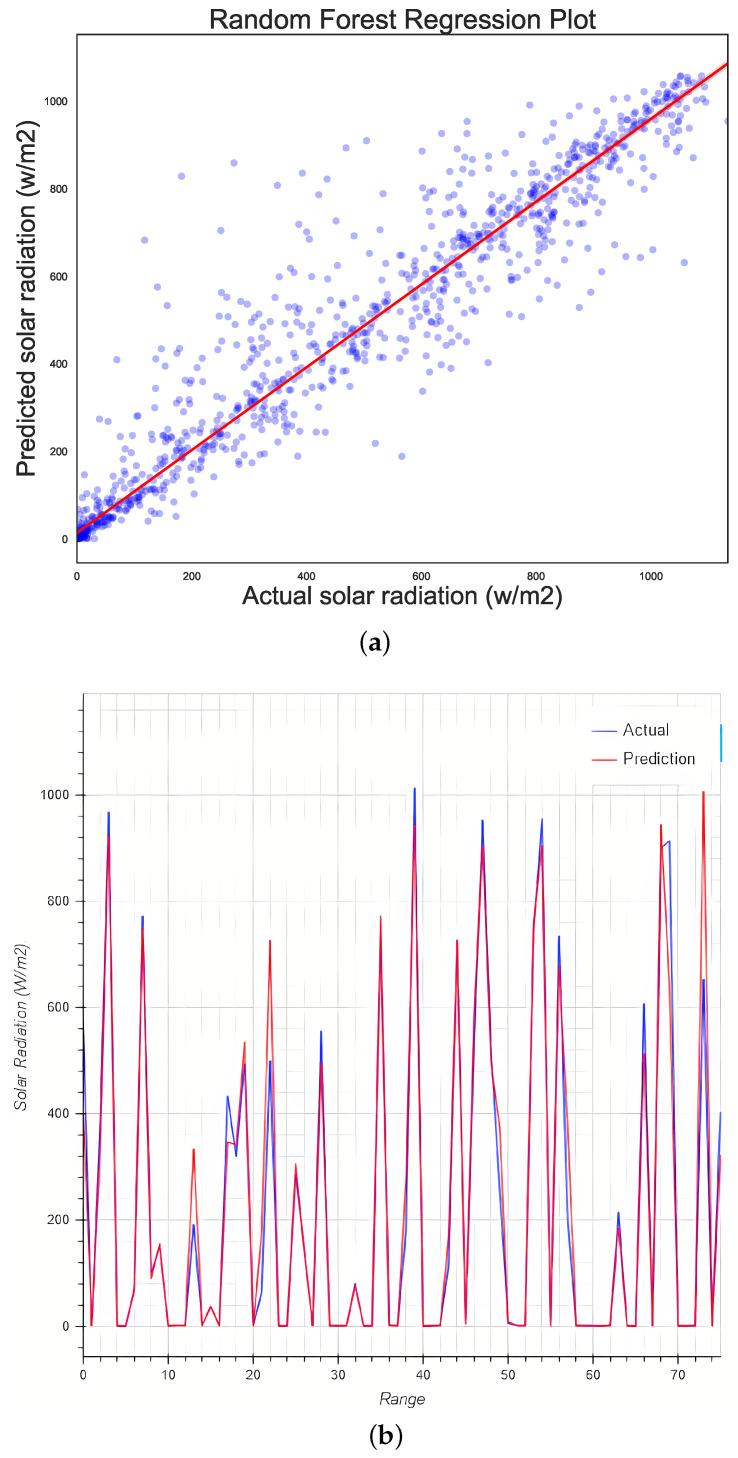
(**a**) Regression plot and (**b**) model validation plot for the RF algorithm.

**Figure 12 micromachines-13-01406-f012:**
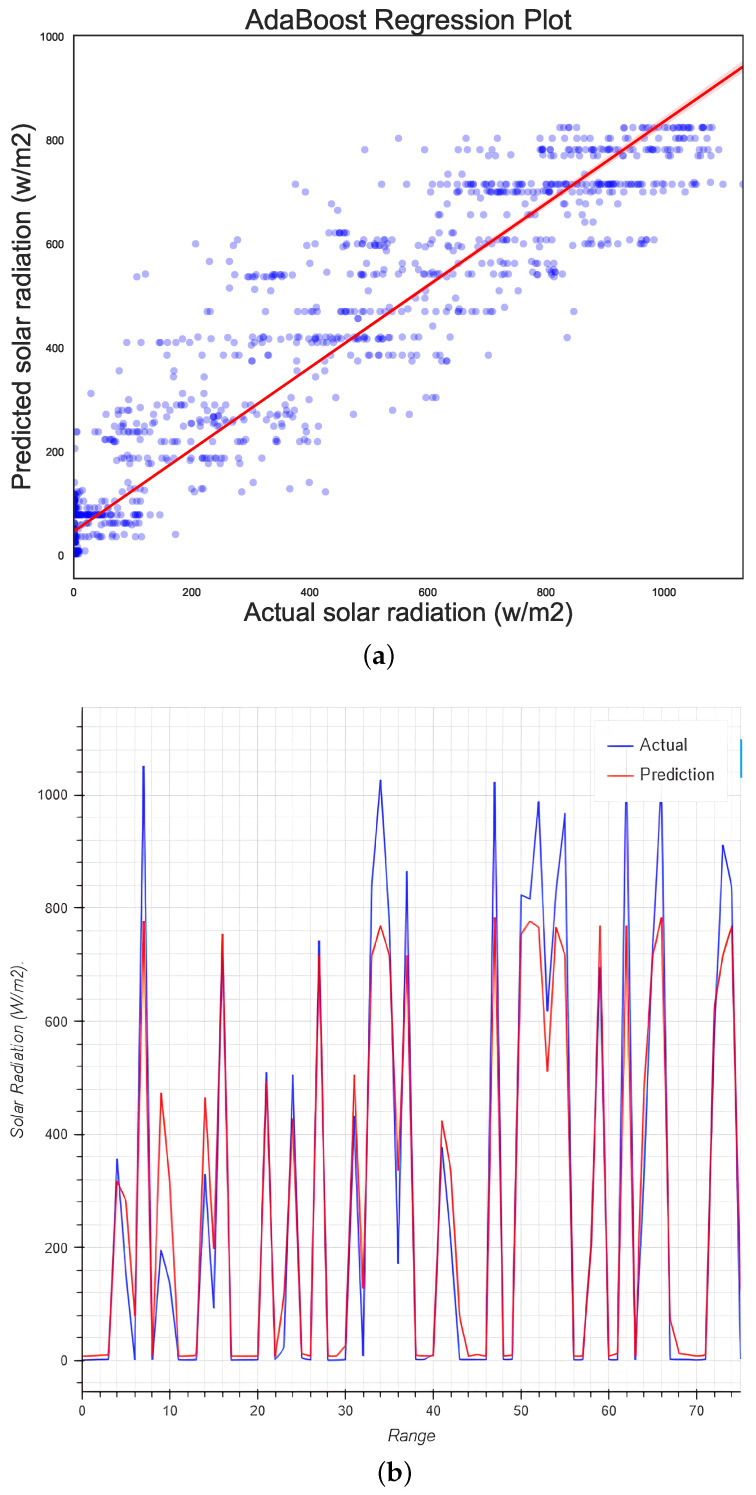
Regression plots for (**a**) Adaboost regression library and (**b**) the predicted results vs. the real data.

**Figure 13 micromachines-13-01406-f013:**
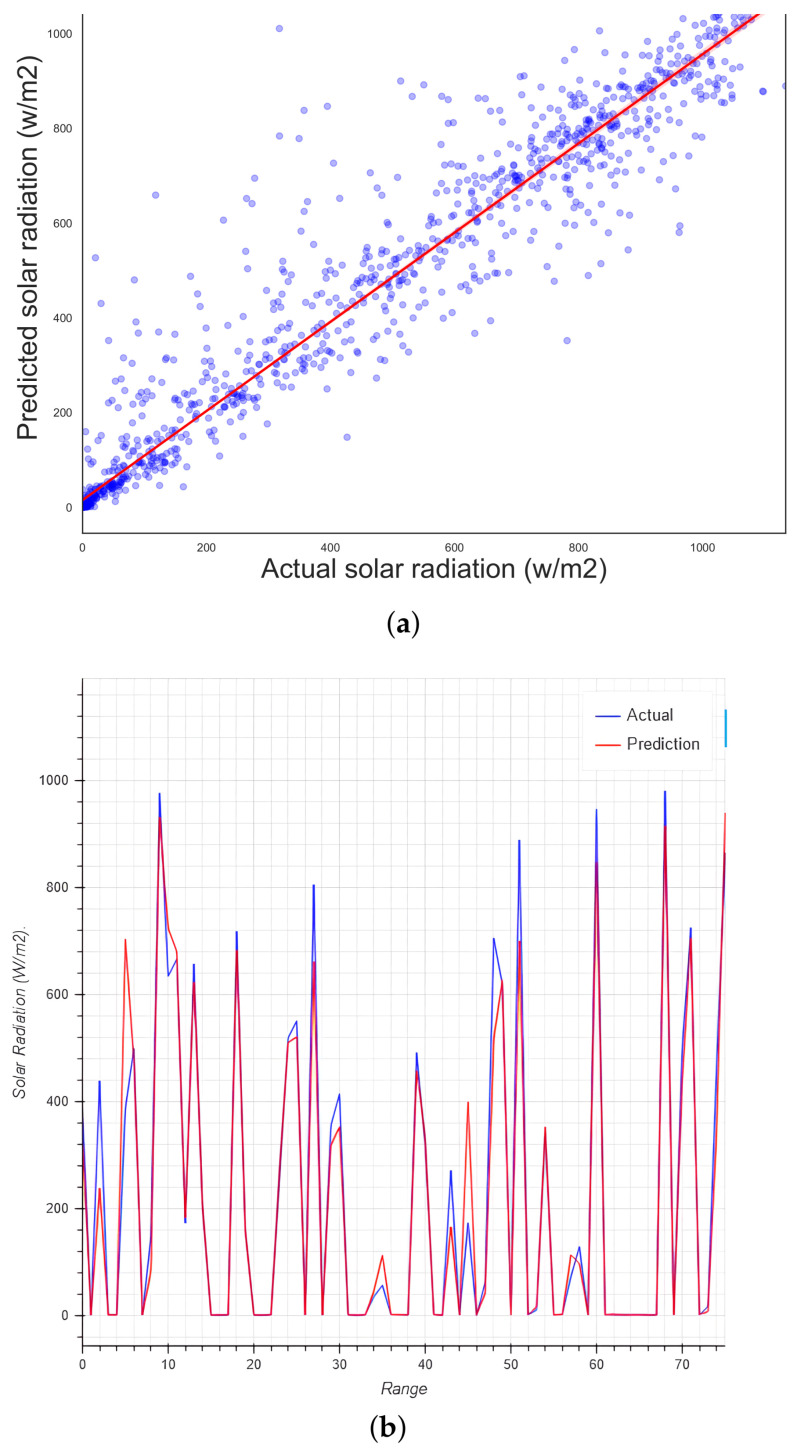
(**a**) Regression plot for the method proposed in Algorithm 1 and (**b**) validation plot for the method proposed in Algorithm 1.

**Table 2 micromachines-13-01406-t002:** Results for correlation of distance between SR and temperature, humidity, and pressure.

Distance Correlation	Values
Temperature vs. SR	0.557138
Pressure vs. SR	0.063914
Humidity vs. SR	0.459387473802509

**Table 3 micromachines-13-01406-t003:** Results obtained after determining the best features for the RF algorithm.

Parameters	Values
estimators	500
features	4
leaf samples	2

**Table 4 micromachines-13-01406-t004:** Results obtained on Juriquilla, Queretaro dataset for the year 2020.

Model	Mean Square Error (MSE)	Root Mean Square Error (RMSE)	Mean Absolute Error (MAE)	R${}^{2}$	MAPE	Accuracy %
Linear regression	51,603.39	227.16	186.96	0.55	3.045478$\;\times \;10^{15}$	55.48
RNN Regression	52,577.32	229.29	180.48	0.53	4.286	53.96
Support Vector Machine	184,469.84	429.499	264.49	−0.51	9.440542$\;\times \;10^{13}$	−51.46
Random Forest	6184.61	78.64	34.44	0.907	90.52	90.34
Adaboost	12,899.40	113.57	74.69	0.89	2.597892$\;\times \;10^{14}$	89.13
Proposed Optimization (RF)	5799.60	76.15	34.41	0.94	2.322810$\;\times \;10^{14}$	94.68
Proposed Optimization (Adaboost)	19,903.65	141.08	81.20	0.82	1.666626$\;\times \;10^{14}$	82.46

**Table 5 micromachines-13-01406-t005:** Results obtained on Juriquilla, Queretaro dataset for the year 2021.

Model	Mean Square Error (MSE)	Root Mean Square Error (RMSE)	Mean Absolute Error (MAE)	R${}^{2}$	MAPE	Accuracy %
Linear regression	52,610.75	229.37	191.35	0.56	1.304530 $\;\times \;10^{14}$	56.37
RNN regression	45,120.43	212.41	154.95	0.53	7.47	53.78
Support Vector Machine	129,502.15	359.86	181.60	−0.33	1.316907 $\;\times \;10^{14}$	−33.57
Random Forest	5573.45	74.655	33.638	0.92	2.500177 $\;\times \;10^{14}$	92.54
Adaboost	12,899.40	113.57	74.69	0.89	2.597892 $\;\times \;10^{14}$	89.13
Proposed Optimization (RF)	5011.19	70.78	33.44	0.95	1.596019 $\;\times \;10^{13}$	95.98
Proposed Optimization (Adaboost)	11,715.85	108.239	68.68	0.89	3.832331 $\;\times \;10^{13}$	89.915

**Table 6 micromachines-13-01406-t006:** Results obtained from Hawaii training dataset for the year 2016.

Model	Mean Square Error (MSE)	Root Mean Square Error (RMSE)	Mean Absolute Error (MAE)	R${}^{2}$	MAPE	Accuracy %
Linear Regression	38558.61	196.36	148.77	0.62	41.23	61.27
Random Forest	6601.40	81.249	32.041	0.93	0.2196	93.34
MLP Regression	46466.07	215.55	155.25	0.531	36.93	53.14
Support Vector Machine	137867.73	378.080	203.02	−0.421	0.7810	−42.12
Proposed Optimization (RF)	6170.84	78.554	32.66	0.93	0.49	93.68

**Table 7 micromachines-13-01406-t007:** Comparison of similar works involving solar radiation prediction.

Author	ML Model	Used Variables	Hyperparameter Optimization Method
Our Work	Optimezed RF	SR, Temp, Hg, RH, sunshine hours	Custom Function 1
Chaibi et al. [47]	Optimized RF	Hg, SR, Temp, RH, sunshine fraction	Bayesian Optimization
Zuo et al. [48]	LSTM	Hg, WindSpeed, Temp, RH, Aerosol optical depth	Bayesian Optimization
Sreekumar et al. [49]	SVR	Cloud cover, cloud movement, wind speed and temperature	PSO
Rahman et al. [50]	ANN	SR, Temp, Hg, RH	Modified ADAM
Fuselero et al. [51]	NARX	avg temp, rainfall amount, RH, wind direction, wind speed, and sunshine duration	Gaussian Regression

## Data Availability

The data presented in this study can be freely obtained from UNAM Observatory [43] and Kaggle [52]; aditional data can be obtained from [53].

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
