# Peer review of "Optimized Random Forest for Solar Radiation Prediction Using Sunshine Hours"

_micromachines, 2022, doi:10.3390/mi13091406_

Round 1
Reviewer 1 Report (New Reviewer)
The following are comments for the authors.
1. Lines 9-11: the content of these sentences is not clear.
2. Line 21: please change the word to English: el niño or la niña
3. It is not suitable to locate the sentence of lines 60-61 in the introduction part.
4. Please do more literature on the application of Random Forest (RF), and add the explanation of why the author used RF for Solar Radiation Prediction instead of other machine learning algorithms.
5. Lines 68-75: these lines are not necessary.
6. Figure 2: Please check the left figure, there are two test datasets
7. Figure 5: Correlation matrix plot for the all the weather variables: in fact that the R correlation analysis here seems to be Pearson correlation coefficient, which can only measure the linear relationship between two variables and this is also the disadvantage of Pearson correlation coefficient. To measure the relationship between variables, please try some other metrics, such as Distance Correlation, Mutual information, and the Maximal information coefficient, etc. Please add in the feature investigation.
8. Why did the authors use a ratio of 80/20 for the training and testing dataset, and why did the authors only divide it into training and testing (without a validation set)?
9. Did the author conduct the cross-validation using k-fold?
1. Discussion section must be revised, please add discussion and comparison with results of previous works.
1. Locate section 6.1 Future works after the conclusion part.
Author Response
- Lines 9-11: the content of these sentences is not clear.
Answer.- Changed the sentence in this line, Lines 9-11, to better reflect the work performed inside the article. Also, the spelling and typos were corrected.
- Line 21: please change the word to English: el niño or la niña
Answer.- Changed el niño to El Niño-Southern Oscillation; this the correct name for this climatologic phenomenon (Cai, W., Santoso, A., Collins, M., Dewitte, B., Karamperidou, C., Kug, J. S., ... & Zhong, W. (2021). Changing El Niño–Southern Oscillation in a warming climate. Nature Reviews Earth & Environment, 2(9), 628-644.)
- It is not suitable to locate the sentence of lines 60-61 in the introduction part.
Answer.- It has been removed the sentence in those lines and changed lines 62 -64 to better reflect the machine learning methods used in this article.
- Please do more literature on the application of Random Forest (RF), and add the explanation of why the author used RF for Solar Radiation Prediction instead of other machine learning algorithms.
Answer.- It has been removed the text from lines 143 to 165, Added the Pareto principle as a new subsection. Shortened and fused to paragraphs from 3.2 subsections to help the flow of the text.
It has been added a new subsection to talk about the Pareto rule or 80/20 rule for machine learning datasets.
- Lines 68-75: these lines are not necessary.
Answer.- It has been removed those lines.
- Figure 2: Please check the left figure; there are two test datasets
Answer.- It has been displayed in Figure 2 to reflect the training, testing, and validation datasets used for prediction. Also, some typos were fixed.
- Figure 5: Correlation matrix plot for the all the weather variables: in fact that the R correlation analysis here seems to be Pearson correlation coefficient, which can only measure the linear relationship between two variables and this is also the disadvantage of Pearson correlation coefficient. To measure the relationship between variables, please try some other metrics, such as Distance Correlation, Mutual information, and the Maximal information coefficient, etc. Please add in the feature investigation.
Answer.- Thank you for your recommendation. The distance correlation between SR and Temp, SR and pressure, and SR and Humidity was performed in this study by using the library scipy.spatial.
- Why did the authors use a ratio of 80/20 for the training and testing dataset, and why did the authors only divide it into training and testing (without a validation set)?
Answer.- The Pareto principle, also called the 80/20 rule, is a fundamental principle of distribution and is based on the principle that statistically, many activities in the world can be explained in this way. We also considered for this research the method of scaling the data. In a very large data set of more than 38000 solar radiation data measured per minute, in the end, to simplify the results we opted to take out the maximum values of each hour and reformulate a new data set with this scale. The results obtained were very similar for both cases, and to simplify the process remembering that it will be used for a low-resource system we opted to use the 80/20 split.
- Did the author conduct the cross-validation using k-fold?
Answer.- We appreciate your observation. Random forest already has its cross-validation metric, known as the OOB score. Our simple proposal already considers this and selects the best OOB score based on the best parameters for the selected hyperparameters, in this case, the number of features, the number of estimators, and the number of maximum samples. For the other methods included (SVM. RNN, Adaboost, ) we used these libraries: cross_val_score, RepeatedStratifiedKFold.
- Discussion section must be revised, please add discussion and comparison with results of previous works.
- Locate section 6.1 Future works after the conclusion part.
Answer.- Thank you for your comment. Future works are now presented in subsection 7.1 as part of the Conclusion section.
Reviewer 2 Report (New Reviewer)
1. Minor spell check is required, for example, in abstract, ‘furher’ should be further
2. Random forest related methods are well known in the field, the section 3 ‘Theoretical Bases’ doesn’t provide any new input to this method, I suggest shortening this section or making it as supplementary materials
3. Section 4.1 ‘study area’ should be in introduction part, also, I suggest shortening the introduction section as well.
4. Figure 4, please fix the typo ‘R^2’
5. Section 5.1 ‘Preparing the data’ should be removed, the basic data structure details should not be presented in the academic manuscript.
6. Line324, 328,the statement ‘ shows that there is a positive linear relationship‘ is wrong, correlation can’t indicate linear relationship, it only investigates whether changes in one variable are associated with changes in other variables. The same wrong statement appears in the other paragraphs as well, authors should check them carefully.
7. ‘It appears that the highest values of SR occur when the ambient pressure is the highest’ could authors explain how they concluded this finding from the heatmap?
8. Line 335-351 all the conclusions are wrong from statistical points of view. For example, in humidity, the authors claim than it is potentially significant, but correlation matrix itself is not allowed to tell whether a variable Is significant or not. Also, in pressure, authors claim that it does not have a direct relationship with RH, which is a wrong statement either because only a simple correlation test only tells the trend of variables but can’t tell the relationship or significance.
9. From the results sections, it shows that authors are not professional in machine learning field, the explanation are not appropriate, for example, in RNN, what does it mean ‘RNN could not properly train data’, what’s the training curve? What’s the training accuracy ? also in SVR, authors indicated that the accuracy score is -51%, it definitely can’t happen when the other models can provide a reasonable prediction.
1 In results, authors implemented poorly on SVR and Neural Nets, but they still suggest using RNN in the further, I’m wondering whether authors understand how to use ML techniques correctly.
1 Also I suggest that authors should include linear regression model as well, not matter how the worse the models can be, they can’t be worse than linear regression.
Author Response
- Minor spell check is required, for example, in abstract, ‘furher’ should be further.
Answer.- We corrected some spelling errors and Changed lines 11-12 to better reflect the work done in the article. The spelling has been checked by a native English speaker.
- Random forest related methods are well known in the field, the section 3 ‘Theoretical Bases’ doesn’t provide any new input to this method, I suggest shortening this section or making it as supplementary materials.
Answer.- It has been deleted from lines 143 to 165. Also, the Pareto principle has been added as a new subsection 3.2. It has been summarized and fused into paragraphs from 3.2 subsections to help the text flow.
- Section 4.1 ‘study area’ should be in the introduction part, also, I suggest shortening the introduction section as well.
Answer.- We removed sections of the introduction that were unimportant to make it more concise when starting to read the article. It has been placed study area to the introduction.
- Figure 4, please fix the typo ‘R^2’
Answer.- Thank you for your comment. We fixed the typos.
- Section 5.1 ‘Preparing the data’ should be removed, the basic data structure details should not be presented in the academic manuscript.
Answer.- Thank you for your comment. Sub-section 5.1 was removed at the suggestion of the revisitor. We added lines 304 to 414.
- Line 324, 328,the statement ‘ shows that there is a positive linear relationship‘ is wrong, correlation can’t indicate linear relationship, it only investigates whether changes in one variable are associated with changes in other variables. The same wrong statement appears in the other paragraphs as well, authors should check them carefully.
Answer.- It has been changed the full sentence in lines 323-328 with: ”With a score of 0.56, the Pearson correlation matrix shows that there is a relationship between solar radiation and temperature, to a less extent to humidity with a score of -0.47, values between -1 and +1 (for example, r = 0.8 or -0.4) indicate that there is variation around the line of best fit, and pressure with a score of -0.04 indicates a little relationship with it, although pressure is strongly related to temperature and humidity.”
- ‘It appears that the highest values of SR occur when the ambient pressure is the highest’ could authors explain how they concluded this finding from the heatmap?
Answer.- Thank you for your concern. This line was deleted.
- Line 335-351 all the conclusions are wrong from statistical points of view. For example, in humidity, the authors claim that it is potentially significant, but correlation matrix itself is not allowed to tell whether a variable Is significant or not. Also, in pressure, authors claim that it does not have a direct relationship with RH, which is a wrong statement either because only a simple correlation test only tells the trend of variables but can’t tell the relationship or significance.
Answer: We made the correlation matrix with Pearson's coefficient and rewrote this paragraph to reflect the new information in Lines 319 to 324. We added to section 5.2 Correlation of distance as a new metric to evaluate the variables temperature, humidity, and pressure vs. solar radiation. The results obtained are as follows.
According to the theory, distance correlation comes in handy because it measures the strength of the association between nonlinear random variables. The distance correlation goes from 0 to 1, where 0 implies independence between X and Y, and one implies that the linear sub-spaces of X and Y are equal. It goes beyond Pearson's correlation because it can detect more than linear associations and work multi-dimensionally.
Distance correlation Values
Temp VS SR: 0.557138
Pressure Vs SR: 0.063914
Humidity Vs SR: 0.459387473802509
It is possible to observe a non-linear relationship between solar radiation and temperature, and the same is true for humidity; pressure does not seem to have a direct impact on solar radiation.
- From the results sections, it shows that authors are not professional in machine learning field, the explanation are not appropriate, for example, in RNN, what does it mean ‘RNN could not properly train data’, what’s the training curve? What’s the training accuracy ? also in SVR, authors indicated that the accuracy score is -51%, it definitely can’t happen when the other models can provide a reasonable prediction.
Answer.- Thank you for your observation. The Result section has been restructured from 336 to 338. Then, the linear regression model has been added to present a point of comparison.
For the SVM regression model, we analyzed other input parameters to try to obtain such as different kernels RBF, Lineal, and polynomial, but those only increased the computed time significantly of training but with the same poor performance.
1 In results, authors implemented poorly on SVR and Neural Nets, but they still suggest using RNN in the further, I’m wondering whether authors understand how to use ML techniques correctly.
These results were obtained using the standard input hyperparameters for all the methods. We understand that optimization of ML algorithms for a specific data set could be done. The purpose of this article is to present a simple programming function for tree-based algorithms (RF, Adabost) that improves the compute time (ex., using it for an embedded system) and, at the same time, provides good results up to 95% in accuracy for prediction. A specific algorithm could be created to have further better accuracy. Nevertheless, that requires having a better understanding of the climatological variables. Again the dataset we are using contains standard features logged by public weather stations; those can be: temperature, rain, wind speed, solar radiation, atmospheric pressure, time of sunrise, and time of sunset.
1 Also I suggest that authors should include linear regression models as well, no matter how the worse the models can be, they can’t be worse than linear regression.
Answer.- Thanks for the suggestion. We added the linear regression models in subsection 5.4
Round 2
Reviewer 1 Report (New Reviewer)
The authors have revised the paper but the following comment in the last review has not been addressed. Please kindly revise it.
- Discussion section must be revised, please add discussion and comparison with results of previous works.
Author Response
- Discussion section must be revised, please add discussion and comparison with results of previous works.
Answer.- Thank you for your observation. In order to discuss and compare our contribution, we added a table (Table 7) that compares our work with the reported in the state of the art. Also, the following paragraph is added in the Discussion section, lines 436-442 and 443-451:
Although multiple works are related to improving the performance of predicting models, this work is distinguished by presenting two main contributions. The first contribution is that the proposed model considers the sunshine hours variable that affects the quality of the solar radiation measured in a certain zone. This variable can be calculated using the public data from the automated weather stations. The second contribution is that the optimized RF algorithm that throws the best hyperparameters obtained by Algorithm 1 presented in this work. This contribution can be compared and contrasted according to Table 7.
Reviewer 2 Report (New Reviewer)
concerns addressed
Author Response
thank you
This manuscript is a resubmission of an earlier submission. The following is a list of the peer review reports and author responses from that submission.
Round 1
Reviewer 1 Report
Weaknesses:
(-) There are English issues.
(-) References are inadequate.
(-) The introduction must be improved.
(-) The related work section must be enhanced.
(-) The method is not novel enough.
(-) Some improvements are needed in the description of the method.
==== ENGLISH ====
· The paper has several typos. Authors need to proofread the paper to eliminate all of them.
· Some sentences are too long. Generally, writing short sentences with one idea per sentence is better.
==== FIGURES ====
· The text of some figure(s) is too small. Authors should make sure that the text can be read if printed on paper.
· All figures in the results section are missing the x and y labels.
==== REFERENCES ====
· There are many irrelevant references. Authors should remove them to keep those closely related to the paper's topic.
· The paper should be updated to include more recent references, preferably from the last 2 or 3 years.
==== INTRODUCTION ====
· The introduction should clearly explain the key limitations of prior work that are relevant to this paper.
==== RELATED WORK ====
· The related work section is not well organized. Authors must try to categorize the papers and logically present them.
· The authors should clearly explain the differences between the prior work and the solution presented in this paper.
· No need to add an empirical model table. Just compare with the latest ML models
· Add more latest references only. And remove old references
· Instead of writing "reference [22]" write author et al.
==== METHOD ====
· A novel solution is presented but it is important to better explain the design decisions (e.g. why the solution is designed like that)
· It is important to clearly explain what is new and what is not in the proposed solution. If some parts are identical, they should be appropriately cited, and differences should be highlighted.
· Explain the variables used in Algorithm 1? What is n,f,s, and other variables represented in this algorithm?
· What is the need for Algorithm 2? Procedure PREP_DATA is initializing the variables which are supposed to be read from the data file (CSV or Excel). And do you really think that you need a procedure to delete just one column (DELETE_SUNTIMES), and sort?
==== RESULTS ====
· No need for Algorithms 3, as it is not explaining anything to the readers. It is just a piece of code, not the algorithm.
· You have added evaluation metrics in section 3.6, remove Algorithms 4
· Table 1 caption is about null values. How do the readings in the table show null values?
· Table 2 shows that optimal features are 4, name them in the manuscript
· Why MAPE is missing for some models?
Reviewer 2 Report
The article is out of the scope of Micromachines journal
Reviewer 3 Report
Knowing exactly how much solar radiation is reaching a particular area is helpful when planning solar energy installations. This study is significant for developing renewable energies. However, the method in this study is a very traditional method. Section 3 is redundant and the improvement described in Section 3.5 is very simple. I do not find any real innovations in this study. In Result section, linear regression, support vector machine and neural networks are very old methods, some up-to-date methods should be used to conduct experiments.
Reviewer 4 Report
Present study has explored the impacts of solar radiation prediction using optimized random forest. The present study shows that more than 90% of accuracy . I suggest for the minor revision provided authors should address following comments.
Minor comments
1) Add flow chart of whole analysis
2) Crearly divide training and prediction dataset